# The Role of Additives in Soil-Cement Subjected to Wetting-Drying Cycles

Yulian Firmana Arifin [1,*], Eka Agustina [2], Fransius Andhi [2] and Setianto Samingan Agus [3]

1   Civil Engineering Study Program, University of Lambung Mangkurat, Jl. A. Yani km 35, Banjarbaru 70714, Indonesia
2   Civil Engineering Master Program, University of Lambung Mangkurat, Jl. Brigjen. H. Hasan Basri, Banjarmasin 70123, Indonesia; h2a512011@mhs.ulm.ac.id (E.A.); h2a512012@mhs.ulm.ac.id (F.A.)
3   Mott MacDonald Pte. Ltd., Singapore 239693, Singapore; samingan.agus@mottmac.com
*   Correspondence: y.arifin@ulm.ac.id; Tel.: +62-5114773858

**Abstract:** This study aimed to explore the use of additives in soil–cement mixtures that have undergone a wetting-drying cycle. In total, two types of soil were used, granitic and lateritic, which are widely used in road base construction in the Katingan area, Central Kalimantan, Indonesia. The cement used was the ordinary Portland type I, while the additive utilized was for commercial purposes, and predominantly contained $CaCl_2$. This research was conducted by testing the optimum cement content for each soil to determine the shear strength according to Indonesian standards (i.e., minimum Unconfined Compressive Strength of 2400 kPa). The optimum cement contents of granitic and lateritic soils were deduced to be 5.5% and 5% on a dry weight basis, respectively. The utilization of 0.8% additive resulted in a 0.5% reduction in the optimum cement content of granite-like soil. The results showed that the optimum additive content for granitic soil was higher than that without supplementation, while for lateritic, no changes occurred. The advantage of using supplements, however, was more pronounced in the samples when they had been subjected to wetting–drying cycles. Additionally, at the optimum additive level, the moisture content and soil-cement loss during wetting was always lower than without supplements.

**Keywords:** lateritic soil; granitic soil; additive; soil stabilization; soil-cement





## 1. Introduction

Central Kalimantan is a province in Indonesia which is famous for its vast swampy areas; thus, it is difficult to source granular material for road foundations. Therefore, a soil–cement base is often used as an alternative.

The reliability and performance of this mixture have been widely studied [1–12]. Sunitsakul et al. [1] reported that the shear strength of a mixture is strongly affected by the water–cement ratio, independent of its dry density. The dry density of the compacted mix should be higher than 95% of the maximum dry density of the modified Proctor compaction, as one of the criteria for road base application [1]. In addition, the percentage of cement is directly proportional to the shear strength of the soil–cement base [2,7,8]. This is because, with the increase in cement, the amount of calcium silicate hydrate (C-S-H), calcium aluminum hydrate (C-A-H), and calcium hydroxide ($Ca(OH)_2$) produced by the mixture's reaction also increases [4,11]. Additionally, the soil–cement shear strength increases with curing time [2,3,5,7,11]. Da et al. [2] reported that a mixture soaked in a higher pH groundwater produced greater strength than those immersed in distilled water. This corresponds with the increase in sample pH with a higher percentage of cement [5]. It can be concluded that the ability to resist stress by the mix is influenced by several factors, such as the water–cement ratio, density, curing time, salt content in the soil, and environmental conditions, particularly water and pH.

The addition of cement also improves the compaction behavior of a mixture in the case of fine-grained soils [7]. The compression index decreases, and the coefficient of consolidation increases, with a higher cement content. It has also been found that the soil pores become smaller, and the structure behaves more robustly with an increasing percentage [7]. $Mg^{2+}$, $SO_4^{2+}$, and $Cl^-$ ions have been discovered in soils with high salt content [11], resulting in the reduction in calcium silicate hydrate (C-S-H) and aluminum hydrate (C-A-H) bonds. Consequently, the strength of the soil–cement mixture is reduced in this case. In addition to its application in road construction, this mixture is also used for other purposes, such as grouting and foundations [6,9].

To improve the strength attainment, soil and cement are normally mixed with some additional components, which are either solid or liquid natural or artificial ingredients. This addition always leads physical or chemical changes in the mixture. The use of additives to increase the shear strength of the soil–cement mixture started in the late 1950s; the researcher [13] used 29 additives, such as dispersants, synthetic resins, waterproofing agents, salts, and alkalis. The addition of 0.5–1.0% supplements, such as sodium carbonate, sodium hydroxide, sodium sulfate, and potassium permanganate, significantly increased the soil–cement shear strength by 150% [13]. Adding more substances beyond this did not result in a significant improvement, and partly resulted in strength reduction, as seen in a case where potassium hydroxide, calcium chloride, and sodium chloride were used.

Using different types of additives, such as acids, enzymatic solutions, and calcium lignosulfonate, Blanck et al. [14] obtained distinct compaction, UCS, swelling, permeability, and surface tension tests for various concentrations. At high proportions of calcium lignosulfonate, the shear strength of the soil–cement mix was lower than that at low concentrations. Lime and rice husk ash were also used as additives to increase the soil's resistance level. Lin et al. [15] added nano-silicon dioxide to a sewage sludge ash–cement mixture to improve its plasticity, shear strength, compression, swelling, and permeability behavior. Adding 2% of this compound to samples at the optimum moisture content produced the highest compressive strength. Aryal et al. [16] used polypropylene fiber to improve the performance of a mix in terms of its wetting–drying and freezing–thawing behavior. It was found out that the soil with 10% cement and 0.5% fiber was able to withstand wetting–drying for up to 12 cycles, based on its percentage loss. Organic fiber such as jute was also used to increase ductility [17]. Garbage, such as ceramic waste and marble dust, were combined with a small amount of cement (i.e., 2%) to produce a sub-base material for rural roads and highways [18]. For different purposes, superplasticizer additives were also used to improve the mixture's performance in grouting, to increase soil injectability and shear strength [19]. It was observed that the mix exhibited different behavior dependent upon the soil type, additive, and its percentage. Therefore, the soil–cement mix and the supplements were first tested according to conditions and designation [13].

Researchers have studied the durability of soil–cement mixtures with additives subjected to wetting–drying cycles [20–23]. França et al. [22] observed the addition of 30% limestone to a soil-cement mixture reduced water absorption and increased its compressive strength. Calcite and gibbsite-rich limestone have also been used in granite waste–cement mixtures. The sample with 60% waste and 5% limestone met the requirements for strength after experiencing wetting–drying cycles for 90 days [20]. De Souza and Lucena [23] replaced water with cassava wastewater, containing calcium and potassium, when making brick soil–cement. After seven days of wetting-drying cycles, the strength, water absorption, and loss of mass of the sample met the established criteria. These results have demonstrated the successful use of additives rich in calcium on soil–cement affected by wetting–drying cycles. The importance of the calcium content in the soil–cement mixture was also reported by Van Ngoc et al. [24]. Deep and rapid damage to soil–cement due to calcium leaching was found in samples submerged in high seawater concentrations [24]. Apart from calcium, fly ash, which contains silica, was also found to reduce mass loss due to wetting–drying processes, with a sample retention strength of 51–88% [21]. Gen-

erally, the mixtures are used for brick. In this case, brushing was not carried out in the wetting–drying test [23].

This article discusses the reliability of two types of soil of predominantly granular material (i.e., granitic and lateritic soils), that have been mixed with cement and commercial additives, with respect to their behavior in wetting–drying cycles. They were chosen because they are widely available in Katingan, where it is not easy to find materials that meet the road base requirements. The most common method is to use a soil–cement mixture from the local soil. This method is more affordable than ordering materials from other regions. High rainfall and tides are often encountered in this location, causing the road to be submerged in several places. Therefore, the soil–cement base becomes degraded, as shown on the Tumbang LahangTumbang Samba-Tumbang Kaman road section, Katingan Regency, Central Kalimantan, as indicated by the arrow in Figure 1a. This is in contrast with the soil–cement conditions where the road was not submerged, as shown in Figure 1b. No visible damage appears to the surface of the soil–cement in the figure. In this study, we aimed to find a solution to the problem by mixing an additive rich in calcium into the soil–cement. This was expected to improve the soil–cement mixture's performance against drying–wetting cycles, as shown by the reduced water absorption and loss of mass.

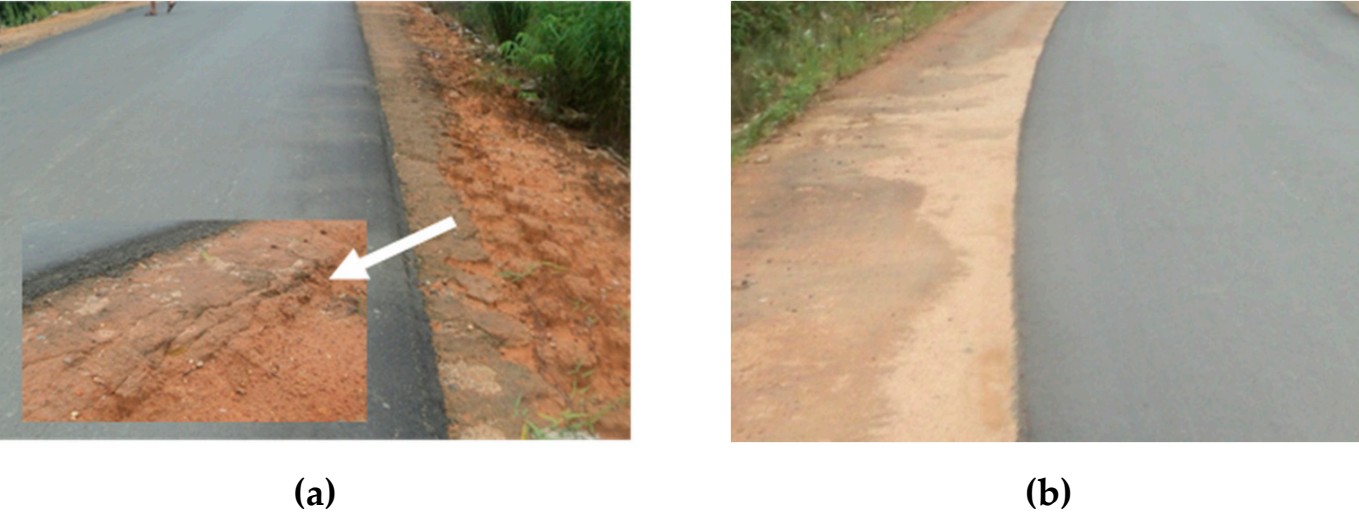

**(a)** **(b)**

**Figure 1.** The appearance of soil-cement as a base (**a**) undergoing wetting–drying cycles, and (**b**) non-submerged road.

## 2. Materials and Methods

### 2.1. Materials

One of the materials used was a granitic soil taken from Hampalit, Katingan Hilir, in Central Kalimantan. The deposits at the location are shown in Figure 2. Another material was a lateritic soil from Tumbang Kaman, about 100 km to the north of the district capital of Katingan, Kasongan, Central Kalimantan. This soil is a type used in road applications, as shown in Figure 1. The basic and engineering properties of the two soils are summarized in Table 1. The two samples had almost the same composition, which predominantly was sand. Both were classified as silty sand (SM) under the Unified Soil Classification System (USCS) [25]. The chemical composition of the granitic and lateritic soils were determined using X-ray fluorescence (XRF) tests, as summarized in Table 2. Although the two samples were classified into the same soil type, the chemical composition of the soils was different. The lateritic soil predominantly contained Si and Fe, while the granitic was largely comprised of Si and Ti. The presence of Si can increase the soil cement's strength by forming C-S-H in the mixture [26].

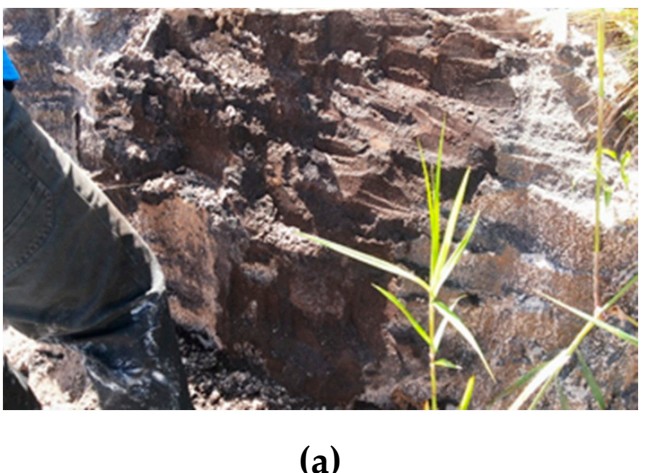
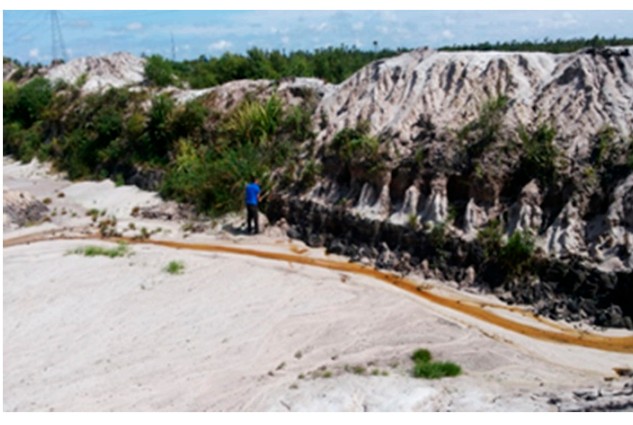

**(a)** **(b)**

**Figure 2.** (**a**) Granitic soil, and (**b**) Granitic soil deposits in Hampalit village, Central Kalimantan.

**Table 1.** Engineering properties of the soils.

| Properties | Granitic | Lateritic |
|---|---|---|
| Specific gravity | 2.64 | 2.64 |
| Water content (%) | 2.4 | 4.3 |
| Gravel (%) | 0.00 | 1.19 |
| Sand (%) | 77.76 | 69.46 |
| Silt (%) | 7.74 | 0.9 |
| Clay (%) | 14.5 | 28.56 |
| Liquid limit (%) | — | 28.59 |
| Plastic limit (%) | — | 22.74 |
| Plasticity index (%) | — | 5.85 |
| Soil Classification (USCS) | Silty sand | Silty sand |
| Unconfined compression strength ($c_u$) (kN/m$^2$) | — | 26.8 |
| Maximum dry density (kN/m$^3$) [1] | 16.33 | 17.73 |
| Optimum moisture content (%) [1] | 12.5 | 14.3 |

[1] Modified Proctor compaction test.

**Table 2.** Chemical composition of soils.

| Composition | Granitic [1] | Lateritic [1] |
|---|---|---|
| Al | 1.77 | 15 |
| Si | 83.12 | 29 |
| Ca | 0.02 | 0.89 |
| Ti | 10.75 | 2.28 |
| Fe | 1.18 | 46.3 |
| Ni | 0.00 | 3.93 |

[1] obtained from the X-ray fluorescence test (XRF).

The cement type used in the study was an ordinary Portland cement type I, with a specific gravity of 3.15. Using the X-ray fluorescence (XRF) test, its chemical contents, as summarized in Table 3, were obtained. The results were comparable with the Portland cement content, which consists of major oxides (i.e., CaO, SiO$_2$, Al$_2$O$_3$, and Fe$_2$O$_3$) and minor oxides (i.e., MgO, SO$_3$, and some alkali oxides (K$_2$O and Na$_2$O)) [27].

**Table 3.** Chemical composition of the cement.

| Compounds | Percentage [1] |
|---|---|
| CaO | 67.28 |
| $SiO_2$ | 18.68 |
| $Al_2O_3$ | 4.30 |
| $Fe_2O_3$ | 4.54 |
| MgO | 1.10 |
| Alkali ($K_2O + Na_2O$) | 1.71 |
| $SO_3$ | 1.28 |

[1] obtained from the X-ray fluorescence test (XRF).

The additive used was a commercial type, which was in the form of a powder. The chemical contents are shown in Table 4, and mainly included chlorine (Cl), calcium (Ca), and potassium (K).

**Table 4.** Chemical composition of the additive.

| Compositions | Percentage [1] |
|---|---|
| Cl | 55.7 |
| K | 4.47 |
| Ca | 37.6 |
| Fe | 0.18 |
| Ni | 0.964 |
| Cu | 0.092 |

[1] obtained from the X-ray fluorescence test (XRF).

*2.2. Methods and Procedures*

Each soil density was achieved by compacting the samples by following the Modified Proctor Standard to obtain the optimum moisture content of the lateritic and granitic samples, which were 14.3% and 12.5%, respectively, with a maximum dry density of 17.73 kN/m$^3$ and 16.33 kN/m$^3$, respectively, as shown in Table 1.

Unconfined compression strength (UCS) tests were carried out on each sample at its optimum moisture content and maximum dry density, with various cement percentages of 4%, 4.5%, 5%, 5.5%, and 6% on a dry weight basis based on SNI03-6887-2002 [28], which was similar to ASTM D-1633-2000 [29]. This test is commonly used to determine the effect of cement on the soil [1–3,5–8,10,11,13–15,17].

Based on the Indonesian standard (SNI03-3438 1994) [30], the optimum cement content is at a UCS of 2200 kPa. Following the latest and more specific standard, the general specification for highways, a UCS of 2000–2400 kPa is required [31]. It should be noted that the required soil shear strength for road applications differs from country to country. Antunes et al. [5] compared the strength required by several countries. Table 5 shows the required mechanical specifications compared to those used in Indonesia; however, in this study, the maximum value was used (i.e., 2400 kPa).

The wetting–drying test was carried out based on the Indonesian standard (SNI 6427 2012) [32]. A No. 4 (4.75 mm) sieve was used. In total, two samples were used in the wetting–drying test. One was used for any changes in absorption (i.e., Specimen No. 1), and the other was for soil loss (i.e., Specimen No. 2). After compaction, the samples were stored in a humid place and protected from free water for seven days. Specimen No. 1 was weighed and measured in dimensions after storage at the end of day 7. Then, the samples were immersed in water at room temperature for 5 h. Specimen No. 1 was again weighed and measured. Both specimens were placed in an oven at 71 °C for 42 h. Then, sample No. 1 was weighed and measured in its dimensions. For Sample No. 2, two firm strokes were given on all areas with the wire scratch brush. It took approximately 18–20 vertical firm strokes to cover the specimen's sides twice, and four strokes on each end. Then, it was weighed. Both samples were re-immersed, and the same procedure was continued

for 12 cycles. At the end of the cycle, the samples were placed in an oven at 110 °C for 24 h to determine the dry weight. This method is similar to the ASTM standard [33]. After 12 cycles, UCS tests were performed to obtain the residual shear strength of each sample. Table 6 presents a summary of the initial conditions of the tested samples. GC and LC refer to granitic and lateritic soils, respectively. The next two numbers indicate the cement and additive content. An additional denotation is given at the end of the sample numbering in Table 6, namely "1" for the volume and moisture change measurements, and "2" is for the soil-cement loss measurements.

**Table 5.** Laboratory UCS required for soil–cement mixtures.

| Layer | U.S. Army Corps for Engineer [5] | German [5] | Portuguese [5] | Southern African [5] | Indonesia [30] | Indonesia [31] |
|---|---|---|---|---|---|---|
| Base | ≥5.17 MPa for 7 days curing time | ≥7.0 MPa for 28 days curing time | Non-specified | 1.5 ≤ UCS ≤ 3.0 MPa for 7 days curing time | 2.2 MPa for 7 days curing time | 2.0 ≤ UCS ≤ 2.4 MPa for 7 days curing time |
| Sub-baseLayer | ≥1.72 MPa for 7 days curing time | ≥0.5 MPa for 28 days curing time | 0.8 ≤ UCS ≤ 1.0 MPa for 28 days curing time | 0.75 ≤ UCS ≤ 1.5 MPa for 7 days curing time | 0.6 MPa for 7 days curing time | Non-specified |

**Table 6.** Initial conditions of the wetting-drying samples.

| Soil | Sample Code | $\gamma_d$ | w (%) | Cement (%) | Additive (%) |
|---|---|---|---|---|---|
| Granitic | GC-5-0-1 | 16.33 | 12.5 | 5 | 0 |
| Granitic | GC-5-0-2 | 16.33 | 12.5 | 5 | 0 |
| Granitic | GC-5-0.8-1 | 16.33 | 12.5 | 5 | 0.8 |
| Granitic | GC-5-0.8-2 | 16.33 | 12.5 | 5 | 0.8 |
| Lateritic | LC-5-0-1 | 17.73 | 14.3 | 5 | 0 |
| Lateritic | LC-5-0-2 | 17.73 | 14.3 | 5 | 0 |
| Lateritic | LC-5-2-1 | 17.73 | 14.3 | 5 | 2.0 |
| Lateritic | LC-5-2-2 | 17.73 | 14.3 | 5 | 2.0 |
| Lateritic | LC-5-5-1 | 17.73 | 14.3 | 5 | 5.0 |
| Lateritic | LC-5-5-2 | 17.73 | 14.3 | 5 | 5.0 |
| Lateritic | LC-5-9-1 | 17.73 | 14.3 | 5 | 9.0 |
| Lateritic | LC-5-9-2 | 17.73 | 14.3 | 5 | 9.0 |
| Lateritic | LC-5-14-1 | 17.73 | 14.3 | 5 | 14.0 |
| Lateritic | LC-5-14-2 | 17.73 | 14.3 | 5 | 14.0 |

In total, two tests were carried out to determine the microscopic samples and chemical components before and after mixing with additives and the wetting–drying processes. The two tests were field-emission scanning electron microscopy (FESEM) and energy-dispersive X-ray spectroscopy (EDAX). Other researchers investigating soil–cement mixes have also used these two methods.

## 3. Results

### 3.1. Optimum Additive and Soil-Cement Content

Figure 3 shows the results of the UCS granitic and lateritic soils. This graph shows that the optimum cement content for both was 5.5% and 5.0%, respectively. The additive content in the mixtures was determined using a trial test by mixing an added component with varying concentrations from 2% to 14% of the soil–cement sample. In the determination of the cement content, the optimum additive percentage produced a sample UCS of 2400 kPa. Its variation with the additive content is shown in Figure 4a,b for the granitic and lateritic soils, respectively. For the granitic soil, lower cement contents (i.e., 4.5% and 5%), with the addition of the same percentage of supplements, were assessed. It was found that the

UCS was still below 2400 kPa. As shown in Figure 4a, the optimum additive content was 0.8% and 6% for 5% cement content. A lower additive content (i.e., 0.8%) was selected and used for further blending. For the lateritic soil (Figure 4b), 2% of the additive was chosen because it gave the required strength (2400 kPa). Although the UCS was almost the same as for the soil–cement mix without additives, its effect on the wetting–drying cycles was easily discernible.

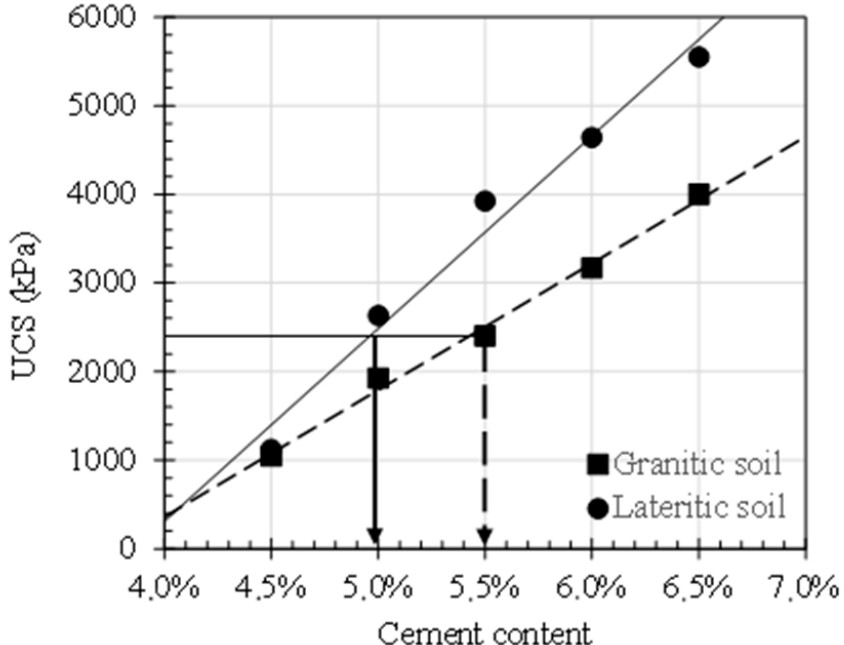

**Figure 3.** Optimum cement content determination.

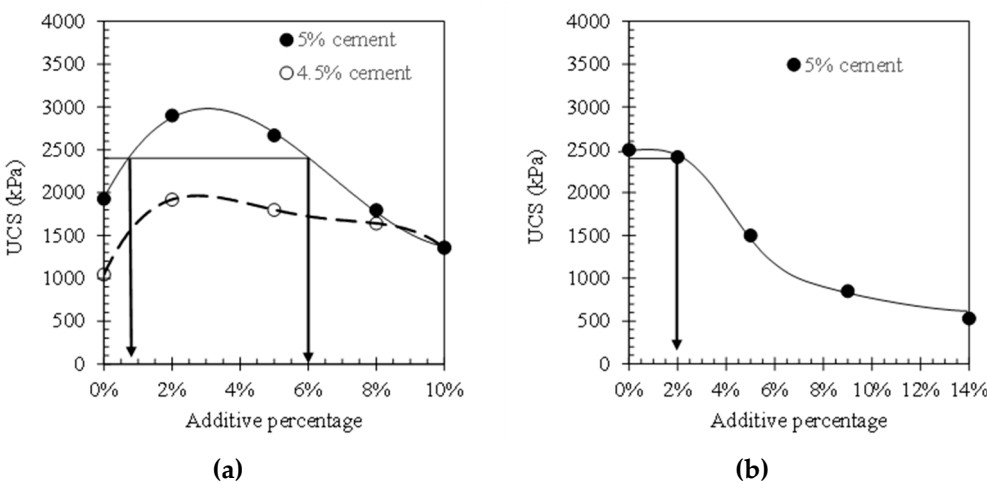

**Figure 4.** Determination of additive percentage in the mixture. (**a**) Granitic soil, and (**b**) lateritic soil.

### 3.2. Granitic Soil

Figure 5 shows the change in water content during the 12 cycles of the wetting–drying process for granitic soil. As shown in Figure 5a, the moisture content of the soil-cement sample after wetting varied by an average of 3.9% for the samples mixed with 0.8% additive, and 14.8% for the samples without it. The addition of 0.8% supplement reduced the amount of water absorbed by the sample by 3.8 times. Meanwhile, for the brushed samples (Figure 5b), the water increased with the number of wetting–drying cycles, which was observed after the sixth cycle. The sample's water

content without additive increased from 16% in the first cycle to 25% in the 12th cycle (a 1.6-fold increase). In addition, with the supplements, it also increased from 4.8% to 20% (or about 4.2 times); nevertheless, the sample water content with additives was still lower than without.

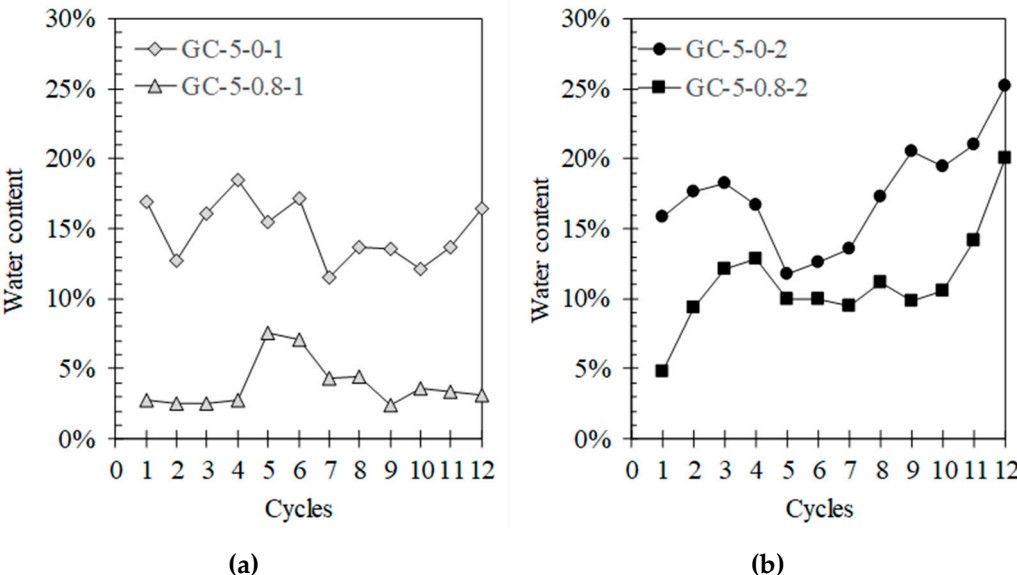

**Figure 5.** Water content alteration throughout the wetting–drying cycles. (**a**) Volume and moisture change specimens, and (**b**) soil–cement loss specimens.

An important conclusion with regards to soil–cement samples that have undergone wetting–drying processes is with respect to the soil–cement loss, which is defined as the ratio of the original calculated sample's oven-dried weight minus its final corrected weight (ASTM D559 1996) [33]. Simply, it is the dry unit weight of the sample per cycle divided by the initial dry density of the sample. Here, the soil–cement loss was shown not only in the brushed samples, but also during soaking (i.e., volume and moisture change specifications). Figure 6a shows that for the soil–cement samples without additives the mixture started losing weight in the second cycle, while for those with supplements this occurred in the third cycle. At the end of the test (i.e., after the 12th cycle), the soil–cement samples without additives exhibited a weight loss of 25% and 17%. The loss for the samples with supplements was 8% less than those without. This was more significant in the sample that was intended for investigation (Figure 6b). The soil–cement loss commenced from the second cycle and increased until the last phase. At the end of the test, the soil–cement loss of the samples without additives was 47%, or 14% greater than those with supplements (i.e., 34%). The addition of these substances reduced the soil–cement loss due to the wetting–drying cycles.

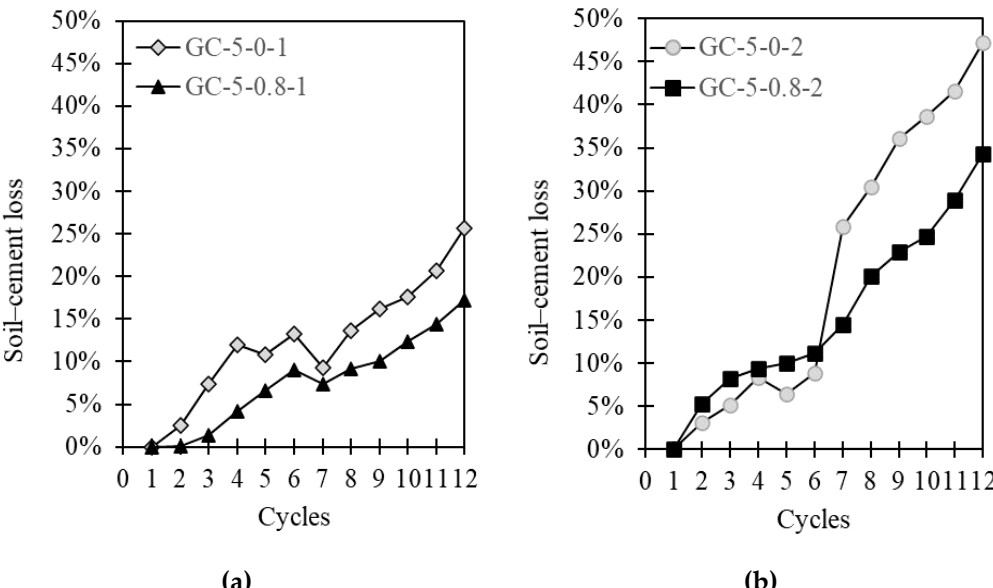

**Figure 6.** Soil–cement loss throughout the wetting–drying cycles. (**a**) Volume and moisture change specimens, and (**b**) soil–cement loss specimens.

Upon completion of these cycles, the samples were tested for their strength (UCS). Sample GC-5-0-2 was not examined due to being broken before testing. Figure 7 depicts the results of the UCS tests on these specimens. Before the wetting–drying cycles, the samples with additives (GC-5-0.8) had a UCS of 2400 kPa, and after the process, it dropped to 1049 kPa for Sample 1 (i.e., for the volume and moisture change measurement) and 678 kPa for Sample 2 (i.e., the specimen for the soil–cement loss measurement). The smallest UCS was observed in the sample without additives (i.e., 441 kPa). It could be concluded that the wetting–drying process decreased the strength of the mixture. Those with additives were twice as strong as those without at the end of the cycles.

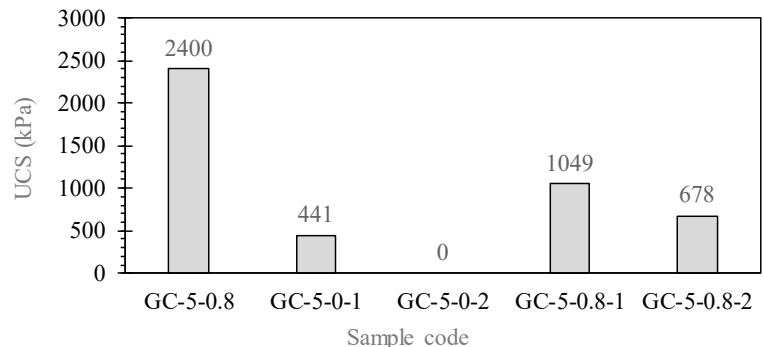

**Figure 7.** Unconfined compression strength of the granitic–cement samples.

Figures 8–10 show the SEM results of the granitic soil samples (Figure 8), the granitic–cement mix specimen (Figure 9), and the soil-cement mix with 0.8% additives (Figure 10). It can be clearly observed in Figure 8a,b that the granitic soil consisted of sand grains and silt particles with irregular shapes and varying sizes, which were smaller than 50 μm. The grains did not appear to bind to one another.

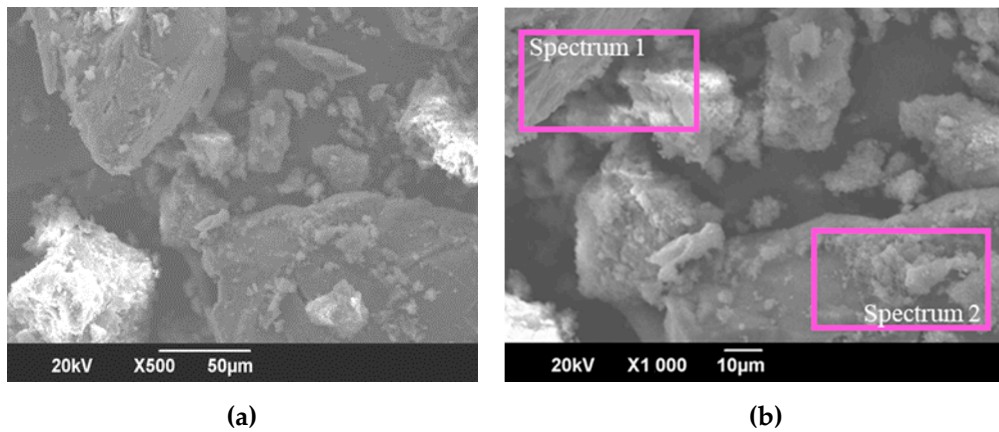

**Figure 8.** SEM Pictures of granitic soil at (**a**) 500× magnification and (**b**) 1000× magnification.

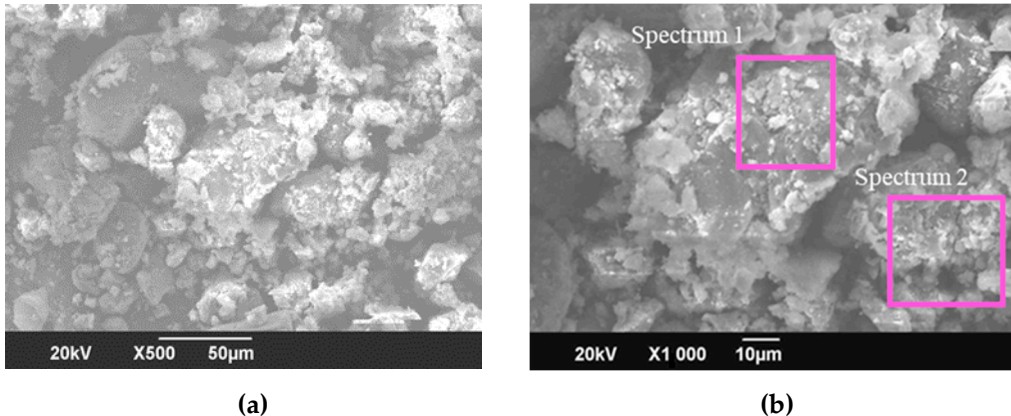

**Figure 9.** SEM pictures of the GC-5-0-1 sample at (**a**) 500× magnification and (**b**) 1000× magnification.

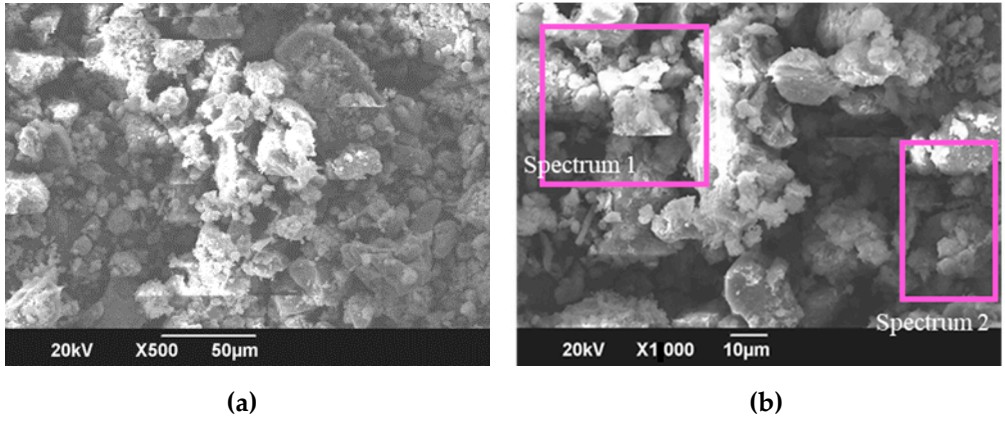

**Figure 10.** SEM pictures of the GC-5-0.8-1 sample at (**a**) 500× magnification and (**b**) 1000× magnification.

Table 7 presents the average chemical contents of this type of soil, extracted with EDX in Spectrums 1 and 2 (Figure 8). This showed the dominance of Si and Ti, confirming the chemical content results from XRF, as shown in Table 2. The addition of cement was observed to produce bonding between the grains, and more compact and smaller pores, as shown in Figure 9. The presence of cement, rich in CaO, was observed from the increase in Ca element at the area where the EDX test was carried out (Figure 9), and the results are shown in Table 7. The Ca content increased to 6.64%.

**Table 7.** Initial condition of the wetting–drying samples.

| Element | Granitic Soil (Figure 8) | GC-5-0-1 (Figure 9) | GC-5-0.8-1 (Figure 10) |
|---|---|---|---|
| Si (%) | 91.95 | 88.82 | 77.06 |
| Al (%) | 1.93 | 1.28 | 6.39 |
| Ca (%) | 0.095 | 6.64 | 15.2 |
| Ti (%) | 6.73 | 1.41 | 1.69 |

The addition of 0.8% additive resulted in more compact clusters with smaller visible pores, as shown in Figure 10a,b. In Table 7, the Ca content increased to 15.2% due to a high content of $CaCl_2$ in the supplement. The presence of this chemical also increased the Ti content due to reduced mobilization of Ti in the soil by $CaCl_2$ [34,35].

Other elements appeared to have little effect; therefore, the influence of additives was not easily recognizable on the different samples' chemical elements, taken in Spectrums 1 and 2 (Figure 10). The average Ca content increased in the specimens, and the SEM results clearly showed differences in the physical conditions of the samples with additives.

*3.3. Lateritic Soil*

Figure 11a,b show the moisture content of the lateritic-cement samples that were subjected to wetting–drying cycles for volume and moisture changes, and soil–cement loss specimens, respectively. The LC-5-14-1 sample (i.e., that with 14% additives) was not tested after the second cycle because it collapsed. The average water content of the samples LC-5-0-1, LC-5-2-1, LC-5-5-1, and LC-5-9-1 were 9.9%, 2.8%, 9.8%, and 10.5%, respectively. Specimens with 2% additives showed the lowest moisture content. For brushed samples, the volume varied but did not increase. This was different from the granitic–cement samples, which showed increased volume after wetting–drying cycles. The average moisture content of the samples were 11.7%, 5.7%, 12.1%, and 12.9% for LC-5-0-2, LC-5-2-2, LC-5-5-2, and LC-5-9-2, respectively. The water content of the LC-5-14-2 sample was not tested because it collapsed after the second cycle.

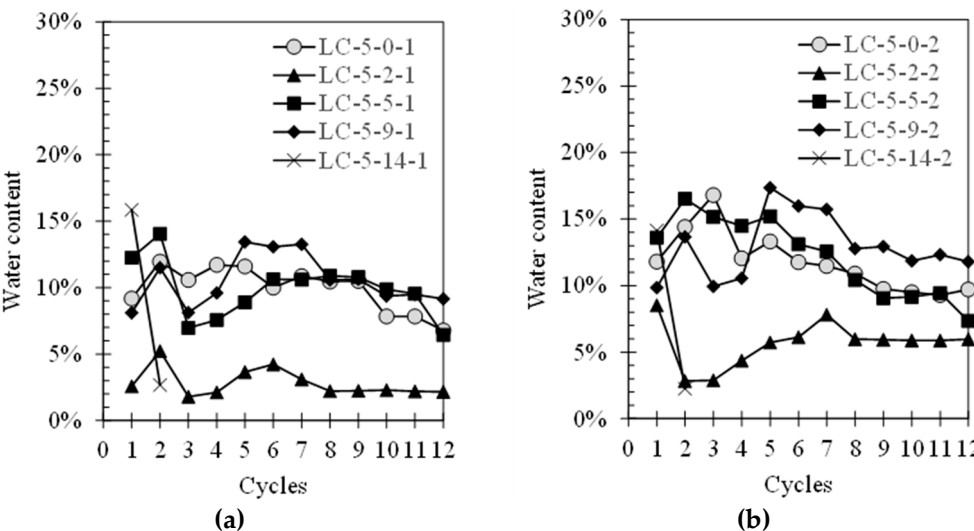

**Figure 11.** Water content alterations throughout the wetting–drying cycles: (**a**) volume and moisture change specimens, and (**b**) soil–cement loss specimens.

Figure 12a,b show soil–cement loss for volume and moisture change specimens. As observed in Figure 12a, the increase in this property occurred from the first cycle to the fifth. In addition, the sample tended not to lose weight. At the end of the cycle, the soil–cement loss samples LC-5-0-2-1, LC-5-2-1, LC-5-5-1, and LC-5-9-1 were 12.6%, 11.7%, 16.6, and 20%,

respectively. Similar behavior was observed in specimens where the sample lost significant weight from cycles 1 to 5. After this, the increase in sample tonnage loss was not that great. At the end of the wetting–drying cycles, the soil–cement loss samples LC-5-0-2, LC-5-2-2, LC-5-5-2, and LC-5-9-2 were 14.5%, 13.7%, 18.4%, and 21.6%, respectively. These results indicated that the sample experiencing the least weight loss was that with the addition of 2% additives (i.e., LC-5-2) for both tests, as shown in Figure 13. The addition of more than 2% supplements resulted in an increase in soil–cement loss.

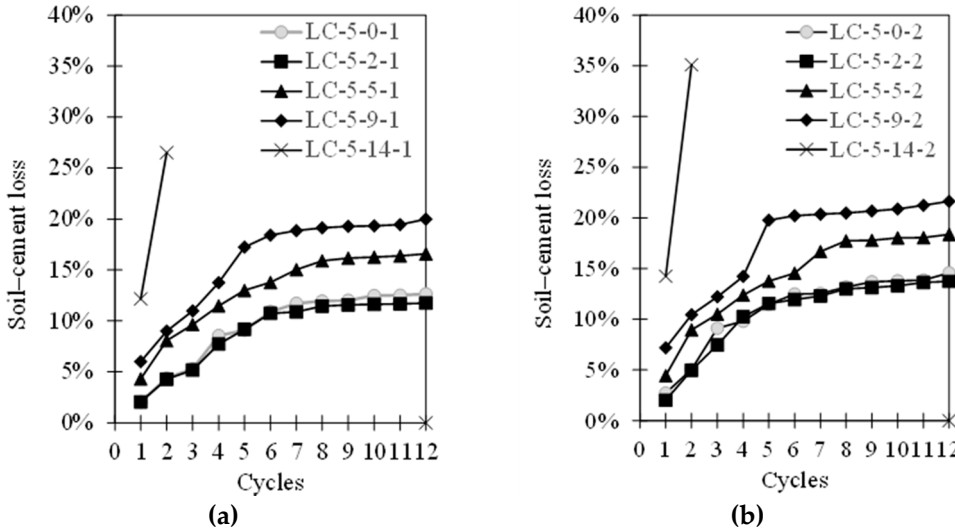

**Figure 12.** Soil–cement loss throughout the wetting–drying cycles: (**a**) volume and moisture change specimens and (**b**) soil–cement loss specimens.

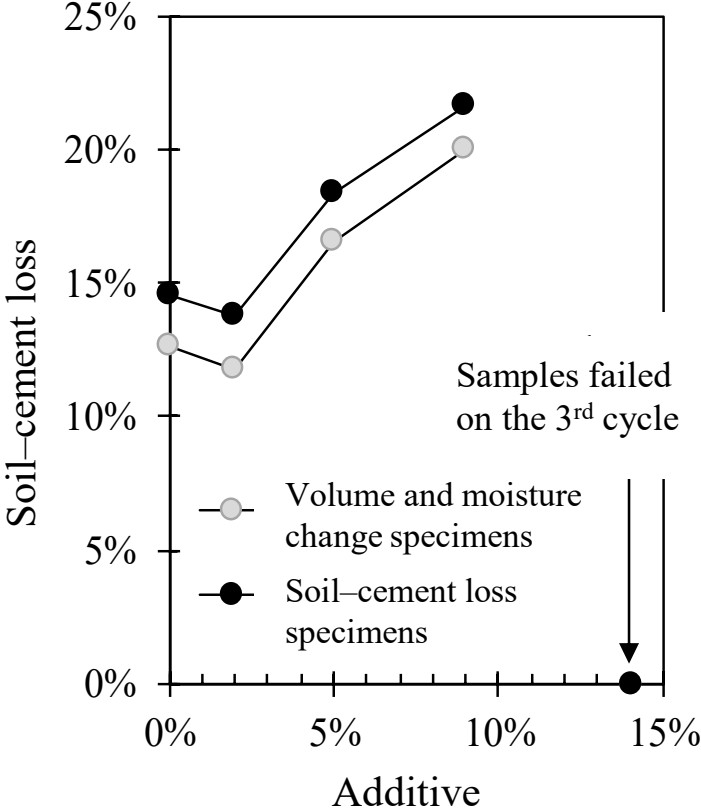

**Figure 13.** Soil–cement loss as a function of the additive content of lateritic soil.

After the wetting-drying test, the samples were examined using UCS, and as shown in Figure 14, the results were compared with UCS specimens before the wetting–drying tests. As observed in Figure 14, this process did not significantly affect the sample UCS, either with or without additives. There was no discernible difference between the two. In addition, the higher the percentage of the additives, the lower the UCS value. These results indicated that the addition of supplements does not always result in a positive trend. Investigations needed to be carried out for each type of soil, and the additives used. These results were in accordance with previous findings [3,13].

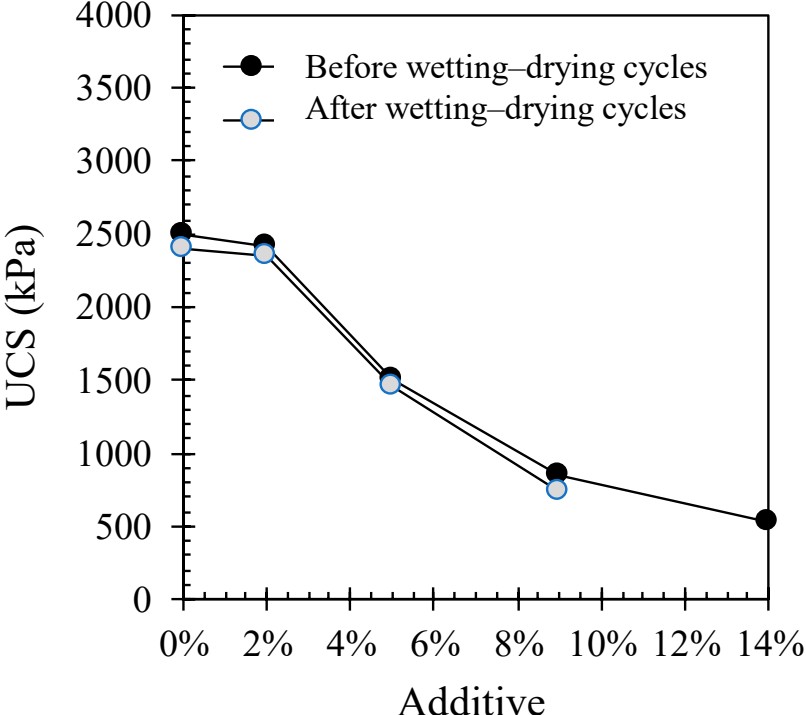

**Figure 14.** UCS as a function of additive content before and after wetting–drying cycles of lateritic soil.

Figure 15 shows SEM photos of samples of lateritic soil (Figure 15a), soil-cement (Figure 15b), and soil–cement-additive mixtures (Figure 15c–f). Figure 15a shows compacted lateritic soil grains with large pores. The granular size varies by even less than 50 μm. The chemical content test was carried out with EDX on Spectrum 1 with the composition shown in Table 8. In the sample, Al, Si, and Fe were the dominant elements, according to the XRF test (Table 2). After adding cement, the specimen was observed to be denser with closed pores, as shown in Figure 15b. Like the granitic soil sample, cement added to the quantity of Ca, which increased from 0.21% to 4.11% in the EDX test results (Table 8).

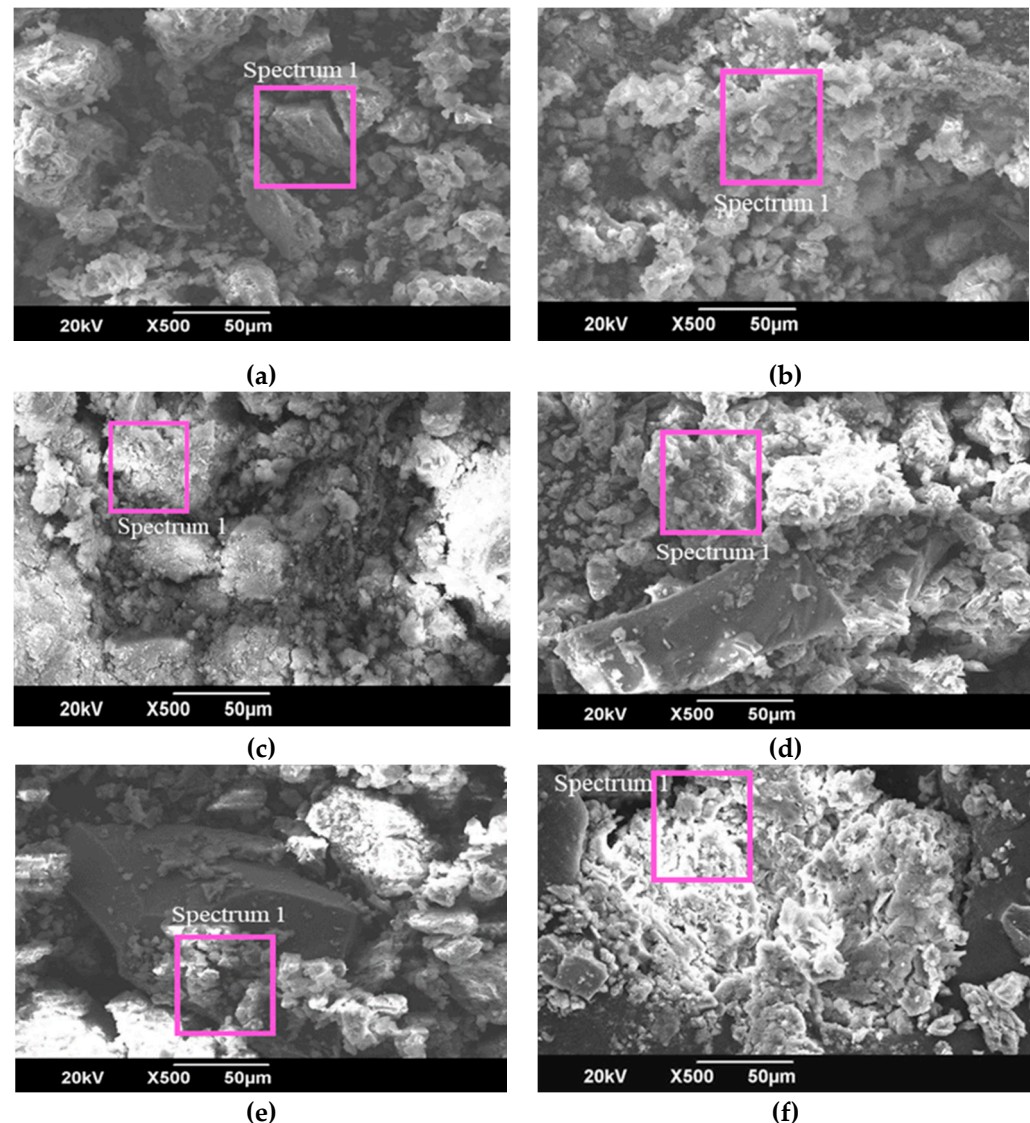

**Figure 15.** SEM photos of samples of lateritic-cement-additive before wetting-drying cycles (**a**) Lateritic soil, (**b**) LC-5-0-1, (**c**) LC-5-2-1, (**d**) LC-5-5-1, (**e**) LC-5-9-1, and (**f**) LC-5-14-1.

**Table 8.** Chemical elements of lateritic–cement–additive mixtures.

| Element | Before Wetting-Drying Process | | | | | | After Wetting-Drying | |
|---|---|---|---|---|---|---|---|---|
| | **Lateritic** **Figure 15a** | **LC-5-0** **Figure 15b** | **LC-5-2** **Figure 15c** | **LC-5-5** **Figure 15d** | **LC-5-9** **Figure 15e** | **LC-5-14** **Figure 15f** | **LC-5-0-1** **Figure 16a** | **LC-5-2-1** **Figure 16b** |
| Al (%) | 31.37 | 30.48 | 34.41 | 28.16 | 30.42 | 26.68 | 32.62 | 36.08 |
| Si (%) | 45.14 | 42.99 | 45.1 | 40.87 | 40.54 | 35.37 | 42.00 | 44.83 |
| Ca (%) | 0.21 | 4.11 | 4.67 | 9.66 | 13.74 | 22.95 | 9.00 | 7.50 |
| Fe (%) | 19.39 | 17.65 | 13.45 | 9.70 | 10.92 | 9.84 | 10.75 | 7.71 |

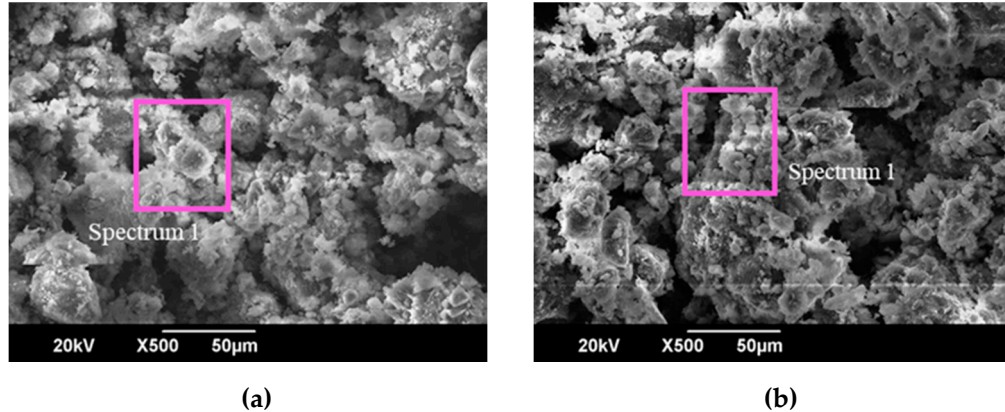

**(a)** **(b)**

**Figure 16.** SEM photos of samples of lateritic-cement-additive after wetting-drying cycles (**a**) LC-5-0-1 After wetting-drying cycles, and (**b**) LC-5-2-1 After wetting-drying cycles.

The addition of 2% additive resulted in a denser sample with even smaller pores. The soil grains were also invisible in this condition (Figure 15c). Excessive supplements caused the cement clusters to reappear; the pores were also clearly visible in this case (Figure 15d–f). The bonds between the cement and soil grains were no longer visible at the additive percentages of 9% and 14% (Figure 15e,f). From the EDX results (Table 8), it was observed that the addition of 2% additives resulted in an increase in Ca, reduction in Fe, and unchanged contents of Si and Al. The addition of Ca, which was supposed to increase the shear strength of the sample, did not occur because of the Fe content reduction. Goldberg [36] reported that iron oxide in clays has a beneficial effect on soil physical properties, increasing its stability and dispersion. Reduced iron oxide content resulted in reduced soil shear strength [37]. When the additive was more than 2%, this resulted in a significant increase in Ca, with the Fe content not changing much, while Si and Al decreased. Iron oxide and aluminum oxide stabilize clay soils by decreasing clay dispersion and water uptake, and increasing micro-aggregation [36], however Fe, Al, and Si's reduced content resulted in reduced soil shear strength [37]. Therefore, it was concluded that additives with high $CaCl_2$ content are not suitable for stabilizing lateritic soils with high Fe content.

Figure 16 shows SEM photos of samples LC-5-0-1 and LC-5-2-1 after the wetting-drying process. It was observed that the two samples showed almost the same conditions; cement clusters with small pores were visible. The two specimens' chemical contents showed that the Al content was slightly increased, and Si remained constant after wetting-drying cycles (Table 8). Meanwhile, the Ca quantity increased due to reduced Fe content in the soil.

## 4. Discussion

The effect of wetting–drying on soil–cement has rarely been examined; therefore, information on reducing its effects is also limited. One strategy is to add polypropylene fiber [16]. In this study, additives rich in $Ca^{2+}$ and $Cl^-$ (Table 4) were used. The addition of $CaCl_2$ to cement is generally used to increase the strength [13,38,39]. The dosage used also varies for different soil types. It was observed that the optimum additive amounts were 0.8% and 2%, corresponding to UCS 2400 kPa, based on the required soil–cement strength standards [31]. The effect of adding more additives than the optimum percentage was also different for the two soils. For lateritic soils, more than 2% supplements resulted in a reduction in the UCS. For granitic–cement, the maximum UCS of 3000 kPa was obtained at an additive content of 3% (Figure 4). This result allowed a reduction in the amount of cement in the mixture, initially of 5.5% (Figure 3). When adding 0.8% additives, the required cement was only 5% (Figure 4). This was due to Si and Al's high content in granitic soil, allowing the formation of more C-S-H and C-A-H. Both compounds play a major role in increasing soil–cement strength [4,11].

Indications of reduced strength due to excess $CaCl_2$ have been submitted by many researchers [38,39] as a consequence of the formation of $3CaO.Al_2O_3.CaCl_2.10H_2O$, due to the presence of $Cl^-$ preventing the formation of C-S-H and C-A-H [4,11]. This effect occurs not only in short-term, but also in long-term strength [4]. The Si and Al content of the two samples tested were different, which resulted in a different effect. The low content of Si and Al in lateritic soils resulted in limited C-S-H and C-A-H formation. The addition of Cl- further reduced their production. SEM results proved that the addition of a Cl-rich additive resulted in a granular shape, which increased with the addition of the additive (Figure 15d–f). This is evidence of the formation of $3CaO.Al_2O_3.CaCl_2.10H_2O$ based on observations made by Xiong et al. [11]. Temperature has also been reported to influence soil–cement [39]. The UCS increased when the sample was kept at 2–21 °C, while the opposite effect occurred when mixing was carried out above 50 °C. In this study, the temperature effect on the increase and reduction in soil–cement-additive strength was neglected, because all tests were carried out at room temperature (between 25–30 °C).

In addition, the discussion around adding additives to soil–cement does not only consider strength, but also the amount of water absorbed and loss of weight due to wetting–drying cycles. The addition of supplements at the optimum percentage (i.e., 0.8% for granitic soils and 2% for lateritic soils) reduced the amount of water absorbed, represented by the samples' low water content, as shown in Figures 5 and 11. The addition of additives resulted in flocculated and clustered structures, as shown in Figure 10a,b and Figure 15c, which increased with higher C-S-H and C-A-H formation [10]. The pores became smaller and denser. Consequently, the water absorbed by the sample when submerged was reduced. The increased strength resulted in weight loss due to soil–cement particle release with less additives, rather than no supplements (Figures 6 and 12). Additionally, the specimens' strength with additives, tested after the wetting-drying cycles, was better than those without (Figure 7).

## 5. Conclusions

The test results of the impact of wetting–drying cycles on soil–cement with additives have been presented and analyzed. Based on the highest compressive strength, the optimum additive contents for the granitic-cement and lateritic-cement mixtures obtained were 0.8% and 2%, respectively. The utilization of additives increased the resistance of the soil–cement mixture in the wetting–drying cycles.

The addition of 0.8% supplements to the granitic soil–cement reduced the amount of water absorbed by the sample by 3.8 times. The soil–cement loss of the samples without additives was 14% greater than those with supplements. For the same soil, the wetting–drying process also decreased the strength of the mixtures. Those with additives were twice as strong than those without at the end of the cycles.

For lateritic soil, the specimens with 2% additive showed the smallest moisture content for both volume change and the soil loss test. Meanwhile, the mass lost due to the wetting–drying process on these soils with additives was slightly smaller than for those without additives. This result was also seen in the residual strength measured after the wetting–drying test. The effect additive was different to that for granitic soil. The chemical content of the soil used affected the success of the additives.

**Author Contributions:** Conceptualization, Y.F.A.; methodology, Y.F.A.; investigation, E.A. and F.A.; writing—original draft preparation, Y.F.A.; writing—review and editing, Y.F.A. and S.S.A.; visualization, E.A. and F.A. All authors have read and agreed to the published version of the manuscript.

**Funding:** This research received no external funding.

**Institutional Review Board Statement:** Not applicable.

**Informed Consent Statement:** Not applicable.

**Conflicts of Interest:** The authors declare no conflict of interest.

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
