# Peer review of "The Role of Additives in Soil-Cement Subjected to Wetting-Drying Cycles"

_infrastructures, doi:10.3390/infrastructures6030048_

Round 1
Reviewer 1 Report
The manuscript is suitable for publication in this journal, however some points must be improved and adjusted for its acceptance and publication:
a) The introduction is insufficient in terms of recent citations and appropriate to the theme. The application of soil-cement involves several characteristics that must be detailed, including works addressing the testing of wetting and drying cycles, I suggest reading and inserting the following works: The durability of soil-cement columns in high sulphate environments; Environmental Durability of Soil-Cement Block Incorporated with Ornamental Stone Waste; Engineering characteristics of compressed earth blocks stabilized with cement and fly ash; Durability of soil-Cement blocks with the incorporation of limestone residues from the processing of marble; Mechanical Properties and Durability of Deep Soil-Cement Column Reinforced by Jute and PVA Fiber; Soil-cement brick with cassava wastewater.
b) The description of the materials is insufficient, the authors must redo the approach, including the description of the wetting and drying cycles;
c) The authors indicate a series of soil characterization results in the methodology section, I suggest switching to the results section, if these are results obtained by the authors;
d) The conclusion is insufficient and does not adequately support the possible results and concussions, a total reformulation is necessary.
Author Response
Dear Editor and Reviewers,
Thank you for the fast response and comments concerning our manuscript entitled “The Role of Additives in Soil-Cement Subjected to Wetting-Drying Cycles”. These comments are all valuable and helpful for improving our manuscript.
According to the reviewers’ comments, we have tried our best to improve the manuscript to meet the journal's requirements. The responses to the reviewers are attached below. The lines listed in the author's response corresponds to the "no markup" condition in the Review menu.
Best regards, Y.F. Arifin
Reviewer’s general comment:
The English spelling and grammar in the paper need to be improved substantially. There are so many errors that is difficult to understand much of paper.
Author's response,
The English of the revised manuscript has been improved and check by a professional English editing. We do thank the reviewer for the valuable comment to improve the quality of the manuscript.
Reviewer’s comments point1:
The introduction is insufficient in terms of recent citations and appropriate to the theme. The application of soil-cement involves several characteristics that must be detailed, including works addressing the testing of wetting and drying cycles, I suggest reading and inserting the following works: The durability of soil-cement columns in high sulphate environments; Environmental Durability of Soil-Cement Block Incorporated with Ornamental Stone Waste; Engineering characteristics of compressed earth blocks stabilized with cement and fly ash; Durability of soil-Cement blocks with the incorporation of limestone residues from the processing of marble; Mechanical Properties and Durability of Deep Soil-Cement Column Reinforced by Jute and PVA Fiber; Soil-cement brick with cassava wastewater.
Author's response point1:
We do thank the reviewer for conscientious review. The introduction section has been modified as suggested by the reviewer in line 85-101. We have added five suggested references. Meanwhile, for one more reference we do not have access to the journal (i.e., Mechanical Properties and Durability of Deep Soil-Cement Column Reinforced by Jute and PVA Fiber). Also, the material used in the literature is fiber as we have not included it in this article.
Reviewer’s comments point 2:
The description of the materials is insufficient, the authors must redo the approach, including the description of the wetting and drying cycles.
Author's response point 2:
The description of the materials and the wetting-drying cycles has been added in the revised manuscript in line 129-133 and line 173-186, respectively, as suggested. We do thank the reviewer for the comment that improve the quality of the manuscript.
Reviewer’s comments point 3:
The authors indicate a series of soil characterization results in the methodology section, I suggest switching to the results section, if these are results obtained by the authors
Author's response point 3:
We agree the suggestion of the reviewer, the sections of line 138-179 (the original manuscript) have been moved in the results section as shown on the revised manuscript in line 201-219.
Reviewer’s comments point 4:
The conclusion is insufficient and does not adequately support the possible results and concussions, a total reformulation is necessary
Author's response point 4:
The conclusion has been reformulated based on the main research finding in line 410-425. We are grateful for the comments from the reviewer.

Reviewer 2 Report
The English spelling and grammar in the paper need to be improved substantially. There are so many errors that is difficult to understand much of the paper. Other comments follow:
- Introduction: The authors should refer to the authors of references when discussing the literature, rather than just the citation itself.
- Introduction: The specific objective of the additive used in the paper is not clear. Is it expected to improve strength, reduce cost, improve wet/dry behavior? There is not a clear objective in the introduction.
- Table 2 and 4: Why don't the XRF compositions include oxygen?
- Line 138-179: These sections may be better in the results section.
- Table 6: Please define all symbols in the table.
- Line 230: "50 mm" Is there a typo in the unit?
Author Response
Dear Editor and Reviewers,
Thank you for the fast response and comments concerning our manuscript entitled “The Role of Additives in Soil-Cement Subjected to Wetting-Drying Cycles”. These comments are all valuable and helpful for improving our manuscript.
According to the reviewers’ comments, we have tried our best to improve the manuscript to meet the journal's requirements. The responses to the reviewers are attached below.
Best regards, Y.F. Arifin
Reviewer’s comments point1:
Introduction: The authors should refer to the authors of references when discussing the literature, rather than just the citation itself.
Author's response point1:
We do thank the reviewer for conscientious review. We already rewrite introduction in line 32-45. Furthermore, we also add more references regarding the effect of wetting-drying cycles on the soil-cement mixture in line 85-101.
Reviewer’s comments point 2:
Introduction: The specific objective of the additive used in the paper is not clear. Is it expected to improve strength, reduce cost, improve wet/dry behavior? There is not a clear objective in the introduction.
Author's response point 2:
The objective in introduction has been revised. The additive was proposed to improve the wet/dry behavior. The sentences have been modified in the revised manuscript in line 102-115.
Reviewer’s comments point 3:
Table 2 and 4: Why don't the XRF compositions include oxygen?
Author's response point 3:
As rightfully stated by the reviewer, the XRF should be possible to include the detection of oxygen element; however, here the light element of oxygen could not be detected in the analysis due to the limitation of XRF equipment's type. We do apologize for this equipment's limitation and thank the reviewer for the remarkable comment.
Reviewer’s comments point 4:
Line 138-179: These sections may be better in the results section.
Author's response point 4:
We agree the reviewer for the suggestion. The sentences have been moved in the section of results in line 201-219.
Reviewer’s comments point 5:
Table 6: Please define all symbols in the table.
Author's response point 5:
The symbols have been defined now in Table 6 in the revised manuscript. The description is also mentioned in line 186-194. This addition is consistent with all symbols used in the text.
Reviewer’s comments point 6:
Line 230: "50 mm" Is there a typo in the unit?
Author's response point 6:
As rightfully stated by the reviewer, there is typo in the unit that should be 50 mm (micro meter). The revised unit can be found in lines 267 and 331. We do thank the reviewer for the correction.

Round 2
Reviewer 1 Report
The authors made the necessary corrections and could be accepted for publication.
Author Response
Dear Reviewers,
We do thank the reviewers for the comment that improve the quality of our manuscript entitled “The Role of Additives in Soil‒Cement Subjected to Wetting‒Drying Cycles”. The English of the revised manuscript has been improved and check by MDPI English service. The revision includes spell errors on the images. We do apologize for sending the manuscript longer than expected time.
We look forward to hearing about our manuscript.
Best regards,
Y. F. Arifin

Reviewer 2 Report
The authors have responded suitably to the reviewer's comments. The English of the paper has been revised and it is more understandable now, but it could still be improved further. However, it is acceptable now.
Author Response

(The authors gave the same response as above.)
